# Body Changes and Decreased Sexual Drive after Dialysis: A Qualitative Study on the Experiences of Women at an Ambulatory Dialysis Unit in Spain

**DOI:** 10.3390/ijerph16173086

**Published:** 2019-08-25

**Authors:** Miriam Álvarez-Villarreal, Juan Francisco Velarde-García, Lourdes Chocarro-Gonzalez, Jorge Pérez-Corrales, Javier Gueita-Rodriguez, Domingo Palacios-Ceña

**Affiliations:** 1Dialysis Unit. Hospital Universitario Infanta Cristina, Avenida 9 de Junio, 2, 28981 Parla, Spain; 2Department of Nursing, Spanish Red Cross, Instituto de Investigación Sanitaria Gregorio Marañón (IiSGM), Universidad Autonoma de Madrid, 28009 Madrid, Spain; 3Palliative Pediatric Care Unit. Hospital Universitario Infantil Niño Jesús, Avenida de Menéndez Pelayo, 65, 28009 Madrid, Spain; 4Department of Physical Therapy, Occupational Therapy, Physical Medicine and Rehabilitation, Research Group of Humanities and Qualitative Research in Health Science of Universidad Rey Juan Carlos (Hum&QRinHS), Avenida Atenas s/n, 28922 Alcorcón, Spain

**Keywords:** chronic renal failure, gender, dialysis, catheter, haemodialysis

## Abstract

Chronic kidney disease (CKD) has considerable effects on the quality of life of patients, impairing everyday activities and leading to lifestyle changes, as well as affecting body image and intimate relationships. Our aim was to describe the experience of female patients with CKD at an ambulatory dialysis unit regarding body changes and sexuality. A qualitative phenomenological study exploring how 18 female patients, treated at the dialysis unit of a Spanish hospital, perceived their sexuality and intimate relationships. Data were collected using in-depth interviews, researcher field notes, and patients’ personal letters. A thematic analysis was performed. Four main themes arose from the data describing the experience of how CKD impacts body changes and sexuality: (a) Accepting body changes, (b) The catheter, the fistula, and body image, (c) Experiencing a different sexuality, and (d) The catheter, the fistula, and sexuality. Patients experienced changes in their body, perceiving it as being bloated or deformed, together with overall decline. The catheter and/or the fistula triggered changes in the way the women dress in an attempt to hide port sites. Women experience sexuality changes, affecting sexual desire and satisfaction. The presence of catheters was found to be the most cumbersome during sexual acts.

## 1. Introduction

Chronic kidney disease (CKD) comprises a group of heterogeneous illnesses which affect kidney structure and function. The diagnosis of CKD is established by the presence of at least three months of glomerular filtration below 60 mL/min/1.73 m^2^ or kidney injury with or without a decrease in filtration [1]. The prevalence of CKD increases with age. In Spain, the prevalence is 9.16%, with 3.3% of the population between 40–64 years old and 21.4% over the age of 64 [2].

In end-stage renal disease (ESRD) patients require renal replacement therapy (RRT) or renal transplants [3]. In the case of RRT, arteriovenous fistulas are used (AF) or permanent venous catheters (PVC), for hemodialysis (HD), or peritoneal dialysis catheters (PDC) leading to changes in body image, and physical and psychosocial demands related with body image which influence sexuality, intimate relationships and self-esteem [4,5].

Up to 10% of the adult population worldwide suffer some degree of CKD, the prevalence of which is higher in women than in men [6]. Additionally, gender differences determine the prevalence of risk factors for the development and progression of CKD [7]. Likewise, women and men show disparities in the clinical presentation of CKD, tolerance to the illness, and response to treatment [8].

The experience of having kidney disease is highly individual, and qualitative research (QR) can provide a more holistic view that may be more meaningful to practitioners. Qualitative methods have been used to study the experiences of patients with CKD regarding the symptoms of the illness itself, the implications of treatment on quality of life, the effects on daily life, and the experience of waiting for renal transplant [9,10,11]. Also, previous studies performed on women have described the process of adaptation to the illness and its treatment, detailing the progression from acceptance of the diagnosis to the need for treatment and the associated adverse effects [12]. The incorporation of spirituality into interdisciplinary healthcare for women with ESRD has been demonstrated to be an essential aspect for living with the illness and the treatment regime [13]. Additionally, previous studies have described the beliefs, values, and experiences of pregnant women with CKD [13]. In contrast, there are no previous qualitative studies describing the experience of female patients related to body changes and sexuality. The purpose of this study was therefore to describe the experience of female patients with CKD on body changes and sexuality.

## 2. Materials and Methods

The guidelines for conducting qualitative studies established by the consolidated criteria for reporting qualitative research (COREQ) [14] and the Standards for Reporting Qualitative Research (SRQR) [15] were followed. Qualitative methods are useful for understanding the beliefs, values, and motivations that underlie individual health behaviors [16].

### 2.1. Design

A qualitative phenomenological study was conducted, following Husserl’s framework addressing the experiences of Spanish CKD female patients. In the field of qualitative studies [16], phenomenology attempts to understand how individuals construct their world view; in other words, it looks through a window into other people’s experiences. The data obtained via qualitative research comes from data collection tools such as interviews, focus groups, and participant observation, and in the form of narrative transcriptions, images (drawings, photography), and documents (diaries, letters) [16]. In the field of qualitative studies [16], phenomenology attempts to identify the essence of participants’ lived experiences (Giorgi, 2005), which is the subjective reflection on human beings when taking part in events in a specific space, and time [17,18]. Husserl’s framework [17] guided this study. For Husserl, the aim of phenomenology is the study of phenomena as they appear, in order to reach an essential understanding of human experience [17,18]. To consider subjective experiences, the researcher assumes a certain attitude of attentive openness and readiness for a proper understanding of the unique meaning of participants’ lived-experiences [18]. This experience always has a meaning for the person who has undergone the same, and thus, phenomenological research uses first-person narratives from the participants themselves as a data source [17].

### 2.2. Research Team

Prior to the study, the researchers’ positioning was established via two briefing sessions addressing the theoretical framework for the study, their beliefs and their motivation for the research [18]. The results of these sessions are shown in Table 1. Six researchers (two women) participated in this study, including two clinical nurses (MAV, JVG), a physiotherapist (JGR), an occupational therapist (JPC), a sociologist (LCG), and a research nurse (DPC). Five of them (DPC, JVG, JPC, JGR, LCG) had experience in qualitative study designs, and had experience in research in health science, and none were involved in clinical activity, nor did they have any prior relationship with the patients included. One researcher (MAV) had clinical experience with chronic kidney disease.

### 2.3. Setting and Patients

The study included CKD patients attending the Ambulatory Dialysis Unit at the hospital belonging to the public health system of Madrid (Spain). The inclusion criteria were: (a) female patients, (b) over the age of 18, (c) patients with a PVC, AF, or peritoneal dialysis catheter in place, (d) diagnosed with chronic kidney disease, following the criteria of Kidney Disease Improving Global Outcomes (KDIGO) [1], and (e) in RRT. The exclusion criteria were: (a) acute kidney disease requiring HD, (b) serious psychiatric or cognitive disorders, (c) inability to communicate in Spanish or provide informed consent.

Purposive sampling was used, based on relevance to the research question (not clinical representativeness) [18,19]. Sampling and data collection was pursued until the researchers achieved information redundancy, at which point no new information emerged from the data analysis [18]. In our study, this situation occurred after including 18 patients.

### 2.4. Recruitment

The researchers were introduced to the patients through the nursing staff at the Dialysis Unit. Previously, the nursing staff had explained the purpose and design of the study to the patients who met the inclusion criteria, at an initial face-to-face contact session during a routine visit. A one-week period was then allowed for patients to decide whether or not they wished to participate. In a second face-to-face session, patients were asked to provide written informed consent and permission to tape the interviews. All the selected patients agreed to participate in the study. There were no dropouts.

### 2.5. Data Collection

Data were collected over a seven-month period between October 2017 and April 2018. The first stage of data collection consisted of unstructured interviews, using open questions, such as: What is your experience with CKD? The second stage consisted of semi-structured interviews based on a question guide (Table 2) in order to obtain information regarding specific topics of interest [14]. The question guide was developed based on accounts obtained from the patients. The interviews were conducted by MAV, DPC, and JVG. The interviews were tape-recorded and transcribed verbatim. A total of 18 interviews were undertaken (one per patient). Overall, 1210 min of interviews were recorded, with 450 min corresponding to the first stage and 760 min corresponding to the second stage. Each of the first-stage interviews lasted between 20 and 40 min (mean duration: 30 min), whereas the second stage interviews lasted between 25 and 60 min (mean duration: 42.5 min). All interviews were conducted at the patients’ homes or in a private hospital room, according to patient preference. Researcher field notes and personal letters provided by patients were also collected. Only one personal letter was obtained from the patients, together with 18 researcher field notes. The researcher field notes provided a rich source of information as participants described their personal experiences, their behavior during data collection, and enabled them to note their reflections concerning methodological aspects of the data collection [16]. During data collection, besides the participants, nobody else was present.

### 2.6. Data Analysis

Complete verbatim transcripts were produced for each of the interviews, researcher field notes, and letters. The texts were collated to enable qualitative analysis [18]. The initial analysis was conducted by MAV, DPC, and JVG. The initial results were subsequently merged in joint sessions, during which data collection and analysis procedures were discussed. In the case of differences of opinion, theme identification was decided by consensus. A systematic text condensation analysis was performed [20]. The process began with the most descriptive content in order to obtain meaningful units. Subsequently, a more in-depth analysis took place by using data reduction in order to classify these into thematic code groups; i.e., grouping of meaningful units referring to the same point or content until the main topics emerged. In this manner, the level of abstraction and complexity of the analysis increased from meaning units to thematic code groups, and finally themes [20]. The final outcome was the identification of themes that represented the female patients’ experiences of suffering CKD. No qualitative software was used on the data.

### 2.7. Rigor

The guidelines established by the COREQ and SRQR were followed [14,15]. Also, the criteria for guaranteeing trustworthiness as cited by Guba and Lincoln were followed [21]. The techniques performed and the application procedures [21,22] used to control trustworthiness are described in Table 3.

### 2.8. Ethical Considerations

The study was approved by the Clinical Research Ethics Committee at Rey Juan Carlos University (code: 2806201711017), and the Clinical Research Ethics Committee at Hospital Universitario Puerta de Hierro Majadahonda (code: 11.17). Informed consent and permission to record the interviews were obtained in every case. Also, the study was conducted in accordance with the principles articulated in the Declaration of Helsinki [23]. Furthermore, we followed the Spanish Biomedical Research Act [24]. Anonymity was ensured by assigning and alphanumeric code to each participant. In addition, no documents or personal information of the participants was shared with people outside the research team.

## 3. Results

Eighteen female patients with CKD were recruited. The mean age of participants was 54.11 years (standard deviation, SD: 14.30), the mean age at diagnosis was 12.5 years (SD: 11.8), and the median time from the beginning of RRT after diagnosis was 6.27 years (SD: 9.57). Their clinical and demographic features are shown in Table 4. Four specific themes emerged from the material analyzed: (a) Accepting body changes, (b) The catheter, the fistula, and body image, (c) Experiencing a different sexuality, and (d) The catheter, the fistula, and sexuality. Table 5 reports some of the female patients’ narratives taken directly from the interviews and personal letters regarding the four emerging themes.

### 3.1. Accepting Body Changes

Women with chronic kidney disease experience significant changes in physical appearance and body image. Some participants felt deformed and bloated, feeling bad about their own body. Women spoke of a turning point, or ‘a before and an after’ marked by when they began treatment, as a result, they felt unrecognizable to themselves in the mirror. However, even with RRT, with the elimination of excess liquid, and associated weight loss, the participants continued to perceive that their image and physique had changed, seeing themselves as very thin, with a body that they could not even recognize. Chronic kidney disease is considered a silent disease, as the body does not produce visible warnings of the illness. Edemas are present in most participants, just before the beginning of RRT. Feeling bloated makes them feel embarrassed and upset. Additionally, CKD affects women’s self-image and self-esteem, altering their own body image, and, as a result, many women feel less attractive.

Most women participating in this study perceived a physical decline similar to premature ageing because of pain associated with more advanced age. Most women were fearful of losing their independence and autonomy, and becoming a burden for their families.

### 3.2. The Catheter, the Fistula, and Body Image

The physical appearance of women with CKD changes throughout the illness, especially after receiving minor surgery procedures to create a vascular access. As a result, they must live with catheters, fistulas, and associated scars, which have a strong influence on their perception of their body image. This implies adapting to living with these devices, without however getting completely accustomed to the same, having to adapt the way they dress. The women in this study used clothing to cover up their body and to hide catheters or fistulas, avoiding cleavages, and having to give explanations.

In participants with AF, the presence of aneurysms and hematomas creates visible marks on their arms, which were perceived as being ugly or ghastly. This leads to questions being asked concerning the fistulas and/or the marks (hematomas) and having to give explanations. Some participants were not affected by this, feeling that the curiosity displayed by other people was normal, however for others, this was accompanied by discomfort and anxiety. Some participants even compared their arms with those of parenteral drug abusers.

### 3.3. Experiencing a Different Sexuality

Most women acknowledged that CKD had affected their sexuality, decreasing their sexual desire, and admitting that they felt less interest or complete disinterest in maintaining sexual relations. They even affirmed that their sexual satisfaction was not the same. Some women associated decreased sexual activity with tiredness, especially after HD. Other participants felt less attractive due to changes in their body image and therefore rejected sexual relations. For other patients, the loss of sexual desire and the lack of sexual relations was due to being of an advanced age, whereas the younger participants did not refer changes in their sexuality due to disease and/or treatment.

The loss of sexual desire and the lack of sexual relations depends on the prior situation. Women who were not sexually active experienced the loss of desire and/or relations as a limitation of the illness itself. In contrast, for women who were more sexually active, this was experienced as a loss, generating sadness and anger, forcing them to maintain sexual relations with no sex drive, and, as a result, less satisfactory. For most women, despite recognizing that the illness causes changes in sexuality, they never sought professional help.

### 3.4. The Catheter, the Fistula, and Sexuality

The presence of devices such as fistulas and catheters influenced the sexual life of the women in this study, generating changes in certain habits or postures during sexual acts, or, on occasion, hampering sexual relations altogether. Some participants described that their partner had more qualms in performing sexual acts than themselves, feeling uncomfortable with the presence of the catheter or fistula. The fistula is the device which, despite affecting the body image, interfered the least with sexuality. In contrast, catheters were those that most affected sexuality and/or those which generated the most difficulties.

## 4. Discussion

Our findings reflect how women with CKD who require RRT experience changes which are negatively perceived due to the presence of physical scars or bulges in the arms [25]. Likewise, previous studies have highlighted skin changes, such as ageing and scaling, together with weight gain and the presence of catheters, especially with HD [26]. In our study, women with ESKD presented feelings of stigma and/or embarrassment related to their new body image, even feeling hatred when seeing themselves in the mirror [4]. Patients must battle daily with body disfigurement and abnormality, having to hide the visible signs of the disease and adapting their clothing [27] or avoiding uncomfortable questions coming from friends and family [28]. In our study, not all women were able to accept and get used to the devices needed for treatment, wishing that they did not have to deal with this in the first place. Thus, the desire for patients to “be normal” regarding their physical appearance reinforces findings from previous studies [26].

The results of this study are in line with previous research concluding that ESRD and its treatment is associated with physical demands that affect the sexuality and quality of life of intimate relations [5]. In ESRD, the physical appearance of the person when fully dressed is often normal, however, the somatic changes are usually hidden underneath the clothing. Thus, the fistulas, and especially the PVC for HD or PCD are visible if the body is uncovered. In physical intimacy, the body is usually exposed, making the physical changes and the presence of devices visible, which makes patients refrain from sexual relations [4]. Sexual dysfunction (SD) is common in all phases of CKD, especially in patients in treatment with HD [29]. In women, this is usually manifested as decreased libido and lubrication, difficulty reaching orgasm, and pain during intercourse [30]. In our study, the main causes of decreased sexual activity in these women are the lack of interest in maintaining sexual relations and tiredness. In recent research, 81% of women were sexually inactive, of which 43% associated this with a lack of interest. Likewise, few women inform of difficulties or dissatisfaction with their sexual life, displaying disinterest for exploring the reasons for the same or consulting treatment options [31].

Previous studies are in line with our findings considering that the lack of sexual activity does not constitute a concern for these women and, therefore, they do not seek professional help [32]. In this study, lack of desire and dissatisfaction with sexual relations is related with the presence of vascular accesses. However, the concerns related with fertility or pregnancy are not addressed, as occurred in other studies [4]. This is probably because most participants were of an age in which maternity was ruled out.

According to the women in our study, being of an advanced age justifies the lack of sexual activity. In contrast, the younger participants did not refer to changes after the diagnosis and the commencement of RRT. In contrast, a previous study by Stewart reported that the decrease of sexual relationships over time does not represent a problem for patients receiving hemodialysis, and that even a time comes when these are interrupted [33].

Another concern expressed by our participants was the lack of independence due to the physical decline experienced, which coincides with other studies in that the insufficiency or incapacity manifested by women hampers the role performed and leads to the loss of identity as a woman [32].

This study has several limitations concerning generalizability. First, we chose to study only female patients. However, gender roles should be taken into account when considering concealment of disease, and gender may underlie differences in disease experience and how it affects daily life, perspectives of pregnancy, fears for maternal, and fetal health, decision-making insecurities and conflicts regarding maternity and autonomy [13]. Therefore, the sexual life changes and experiences of body changes in men with CKD are most likely quite different and should, therefore, be researched specifically. Other limitations are associated with the age of the female patients included in the study as there were no young adult patients, as well as their form of disease, and the site where all the patients were recruited being a dialysis unit. Finally, it is important to consider the perspectives of the patients’ partners for a more comprehensive analysis of the impact of dialysis on sexuality. Indeed, our results cannot be extrapolated to the whole population with CKD, however, they can most likely be applied to other contexts with similar characteristics [16,18].

## 5. Conclusions

Female patients with CKD who require vascular accesses for RRT often experience changes to their body which trigger disorders in their personal body image, as well as changes in their sexual desire and sexual satisfaction. Our results provide insight on how these body changes and sexual life changes are experienced by women with CKD, and may be helpful in dealing with CKD patients during dialysis treatment, follow-up, and discharge. Our study provides a ground to guide further studies addressing the assessment of sexuality, body changes, and quality of life in female patients with CKD. It is necessary to create professional protocols for managing changes in the body and the sexuality of patients. Also, protocols for providing counseling for female patients with CKD should integrate the perspective of the women themselves.

## Figures and Tables

**Table 1 ijerph-16-03086-t001:** The positioning of the researchers**.**

**Theoretical Framework**	Researchers based their approach on an interpretivist paradigm. This paradigm was based on the assumption that human beings construct their own social reality, and that knowledge is built through increasingly nuanced reconstructions of individual experiences.
**Beliefs**	CKD ^1^ and RRT ^2^ provoke changes in women’s sexuality and their body, which impacts upon their private life. Women do not seek help when these aspects of their life are altered and her real perspective is unknown.
**Motivation for the Research**	To gain insight into CKD through the first-hand experience of female patients. To describe and understand female patients’ point of view regarding aspects which they consider relevant to their lives (e.g., sexuality, body changes) and, as a result, improve healthcare services.

^1^ CKD: chronic kidney disease. ^2^ RRT: renal replacement therapy.

**Table 2 ijerph-16-03086-t002:** Semi-structured interview guide.

Investigated Theme	Questions
**Disease**	How would you describe your disease? What does it mean for you? What part of the disease is most relevant to you?
**Treatment**	What do you consider to be the most relevant aspect of the treatment that has been prescribed to you? How much and in what ways does CKD ^1^ treatment restrict you?
**Sexual life**	What are the most relevant changes you have had to make in your sexual life because of CKD? How much and in what ways does CKD restrict you?
**Body**	What changes have you perceived in your body? How do you feel about your body? What is the most relevant aspect for you?
**Devices** **(AF ^2^, PVC ^3^, PDC ^4^)**	How do these devices influence your sexual relations? How do you perceive them in relation to your body? What is the most relevant aspect for you regarding these devices and your body and/or sexuality?
**Social life**	What are the most relevant changes that have taken place in your social and family life? Has your relationship with your wife, friends and close relatives changed because of CKD? If so, in what way?

^1^ CKD: Chronic Kidney Disease. ^2^ AF: arteriovenous fistula. ^3^ PVC: permanent venous catheter. ^4^ PDC: peritoneal dialysis catheter.

**Table 3 ijerph-16-03086-t003:** Trustworthiness criteria applied.

Criteria	Techniques Performed and Application Procedures
**Credibility**	Investigator triangulation: each data source was analyzed. Thereafter, team meetings were performed during which the analyses were compared and themes were identified. Triangulation of data collection methods: including unstructured interviews, semi-structured interviews, and researcher field notes. Participant validation: this consisted of asking the participants to confirm the data obtained at the stages of data collection.
**Transferability**	In-depth descriptions of the study performed, providing details of the characteristics of researchers, participants, contexts, sampling strategies, and the data collection and analysis procedures.
**Dependability**	Audit by an external researcher: an external researcher assessed the study research protocol, focusing on aspects concerning the methods applied and the study design.
**Confirmability**	Investigator triangulation, data collection triangulation. Researcher reflexivity was encouraged via the previous positioning, performance of reflexive reports and by describing the rationale behind the study.

**Table 4 ijerph-16-03086-t004:** Demographic and clinical features of the female participants with chronic kidney disease.

Patient	Age. Years	Diagnosis	Age at Diagnosis. Years	Partner	Treatment Type	Year Treatment Began	Vascular Access Device	Date of Last Vascular Access	History of Vascular Access
1	58	CKD ^1^ secondary to pyelonephritis/interstitial nephritis secondary to ureteral reflux–vesico-ureteral reflux without obstruction	24	Yes	HD ^2^	1984	AF ^3^	09/1999	3 AF
2	63	ESKD ^4^ secondary to chronic bilateral pyelonephritis	38	Separated	HD	1993	AF	01/2016	4 AF
3	33	Lupus nephritis	26	Yes	HD	2011	AF	05/2014	None
4	67	Glomerulonephritis	50	Yes	HD	2001	PVC ^5^	06/2014	Several AF
5	54	CKD secondary to malignant hypertension	50	Yes	HD	2014	PVC	09/2014	2 PVC. 3 AF
6	28	Solitary kidney secondary to left nephrectomy due to clear cell papillary carcinoma	25	Yes	PD ^6^	2015	CPD ^7^	02/2017	None
7	64	Diabetic nephropathy	61	Yes	HD	2015	AF	02/2016	1 PVC
8	32	Lupus nephritis	31	Yes	PD	2017	CPD	05/2017	1 PVC
9	47	Hepatorenal polycystosis	34	Yes	PD	2014	CPD	10/2013	1 PVC
10	50	Chronic tubulointerstitial nephritis, secondary to pyelonephritis and repeated reno-ureteral colic	44	Yes	HD	2017	PVC	10/2017	1 AF
11	56	Renal lithiasis with nephrolithiasis	39	Separated	PD	2017	CPD	09/2017	1 PCD
12	64	Diabetic nephropathy	58	Yes	KT ^8^	2017	None	-	1 PVC. 4 AF
13	46	CKD secondary to malignant arterial hypertension	45	Yes	HD	2017	AF	11/2017	1 PVC
14	69	CKD of unknown etiology Probable nephroangiosclerosis	51	Yes	HD	2000	AF	11/2002	1 AF
15	72	Acute kidney injury AKI 3 Secondary to massive bleeding after epithelioid hemangioendotelioma surgery	67	Yes	HD	2014	AF	01/2015	1 PVC
16	42	Membranous glomerulonephritis	40	Yes	HD	2017	PVC	08/2017	1 PVC
17	51	Polycystic kidney disease	27	No	PD	2018	CPD	10/2017	None
18	78	CKD of unknown etiology	38	Yes	HD	2017	AF	02/2012	None

^1^ CKD: chronic kidney disease. ^2^ HD: hemodialysis. ^3^ AF: arteriovenous fistula. ^4^ ESKD: end-stage kidney disease. ^5^ PVC: permanent venous catheter. ^6^ PD: peritoneal dialysis. ^7^ CPD: catheter peritoneal dialysis. ^8^ KT: kidney transplant.

**Table 5 ijerph-16-03086-t005:** Narratives from female participants with chronic kidney disease**.**

Themes	Patient Narratives
Accepting changes in the body	Changes to their body (deformed, swollen): *“When I have a shower, I see the shape of my body, at first normal, thin, then I see the bulges on the outside and I can’t stand that, I don’t like it. I find that I am different, I find that I am bloated.” (P17, 51 years old, semi-structured interview), “For me, it’s awful, firstly because all my body has become deformed.” (P7, 64 years old, semi-structured interview).* ‘A before and an after’, not recognizing oneself: *“I am not the same as I was before the change, before the illness. Sometimes I feel bad in my own body (…) my body has changed a lot, sometimes I don’t even recognize myself in the mirror.” (P5, 53 years old, semi-structured interview).* An unknown body: *“As a result of slimming, my face looked worse, and I stopped dressing like before, with my high heels, my clothes, the same activity. Then, I went through a really hard time, when I changed physically, it was devastating.” (P2, 63 years old, semi-structured interview), “When I went got in to the hospital I weighed 57 kilos and I was ok, then, suddenly, I have begun to lose weight or they take it off and then, physically I don’t look good to myself. Now I see my arms and I think: ‘Goodness me! I am getting thin to the bone.’ I look at myself in the mirror and I see myself as being really thin.” (P13, 46 years old, semi-structured interview).* A silent illness, edemas (shame and anguish): *“… I couldn’t walk, because of how swollen I was. It was really dreadful and I was embarrassed of people seeing me so swollen.” (P4, 67 years old, semi-structured interview),* “*My belly was getting swollen as if I were pregnant of triplets, due to the liquid. It was rough. I was always crying. People asked me, are you pregnant?” (P16, 42 years, semi-structured interview).* Feeling less attractive: *“I felt that, physically, I had changed, I had gone downhill, I noticed that my husband wasn’t attracted to me. My self-esteem was very low.”* (*P2, 63 years old, semi-structured interview)* Premature ageing: *“My health is getting worse and worse. I am becoming more clumsy, I fall often and that is because the bones are worse. But also, it can be because of the machine, the treatment, the machine eats you up and, of course, you age and get older before.” (P4, 67 years old, semi-structured interview), “I am 47 years old, and, it’s like, all of a sudden I have aged 10 years. Things happen to me that could happen to a 60-year old woman … My bones and my hands ache. I heard my mother complain [like this], but when she was 70, not at my age.” (P9, 47 years old, semi-structured interview),* “*Your body progressively deteriorates. Right now I am 58 and I feel as if I were 80 because I can’t walk due to the pain. Little by little this messes everything up.” (P1, 58 years old, semi-structured interview).* Loss of Independence and autonomy: *“The only thing left now is for me to have another stroke, which is what I fear, and not knowing where I am and what I am doing. Not being able to eat on my own or being unable to get dressed.” (P1, 58 years old, semi-structured interview), “The only thing that I am worried about is causing problems for my daughter and becoming an invalid, not being able to do anything. I am worried about not being able to do things, losing more and more abilities, and doing less”. (P11, 56 years old, semi-structured interview).*
The catheter, the fistula and body image.	Living with catheters, fistulas and scars: *“It’s not easy to see yourself as looking good with a catheter [permanent venous catheter] hanging from your body or with a bulge in your arm [fistula], you can’t even believe it’s your body.” (P16, 42 years old, semi-structured interview),* “*Yes, I look weird, I see myself, and, I don’t know, I think people must not like seeing it [permanent venous catheter]. I can’t avoid it…” (P10, 50 years old, semi-structured interview).* Adapting the way one dresses: *“Getting dressed is an ordeal. In winter it’s no problem, you are very much covered up. In the summer, you don’t wear certain clothes because the catheter can be seen [permanent venous catheter].” (P13, 46 years old, semi-structured interview), “I always cover up on top, before, I used to like wearing a cleavage, but not anymore. Now I am always very covered up. My style of clothes has changed very much. Before I dressed very “provocative”, but not anymore.” (P10, 50 years old, semi-structured interview).* Arteriovenous fistulas (marked and deformed arms): *“When I began dialysis I got many hematomas and such, and isn’t nice, it isn’t nice to see an arm like that, but the fistula is part of the illness.” (P2, 63 years old, semi-structured interview).* Questions about the fistula and/or the marks (hematomas): *“People ask you what happened, what is this or that, but later, you learn to live with it and now I don’t care. I explain about the fistula, and I even tell them to touch it. This is the way I normalized it all, now I am not so self-conscious or anything.” (P3, 33 years old, semi-structured interview), “For me, the fistula is part of the illness you have. It isn’t nice, I don’t like having it, but it doesn’t cause me any complexes. There are people who look at me, as if to say: ‘what do you have on your arm?’, or ‘she looks like a drug-addict’ (…) for others, the bulges look like something abnormal. The bulges that arise from the fistula.“ (P2, 63 years old, semi-structured interview), “For me, the arm is horrible, and all my body is deformed. At the beginning I did hide it, but now I don’t, I know my arm is ugly, what can I do, there’s nothing I can do about it, it’s what I have.” (P7, 64 years old, semi-structured interview)*
Experiencing another sexuality	Less sexual desire and lack of interest: *“You do notice, you lose your sex drive. Yes, that changed for me, I lost my sex drive, and that also affects sexual relations. These aren’t like they were before.” (P2, 63 years old, semi-structured interview), “Because of the illness, I haven’t been able to function properly, it’s not that I haven’t functioned as a woman, I have done so all my life, and I continue to do so, but now I don’t get any satisfaction.” (P14, 69 years old, semi-structured interview)* Decreasing sexual activity because of feeling tired: “*My sexual appetite was decreased, of course, we do have sex, but it isn’t like before, because, like I am saying, I have no sexual desire, especially after the sessions”.(P9, 47 years old, semi-structured interview), “Perhaps I have less but because, when I get out of here [hemodialysis unit] I am like, more tired, and that day I really don’t feel like it.” (P13, 46 years old, semi-structured interview*) Rejecting sexual relations due to body image: “*I don’t have any desire for sex. If you don’t feel comfortable with yourself, it is impossible to relax because you aren’t comfortable, and if he touches you, you cover up. Yes it changes, it changes a lot. I have always been very vain, and now I am unable to take my jumper off in front of him.”* (*P10, 50 years old, semi-structured interview)* Changes in sexual desire and sexual relations due to age: *“When you are young, you fancy everything, but when it’s been a while, that goes [sexual desire], and it becomes less and less.” (P4, 67 years old, semi-structured interview), “When I have intercourse, nothing has affected me, I don’t have any problem.” (P6, 28 years old, semi-structured interview)* Sexual relations without sexual desire and less satisfactory: *“Often, I maintained sexual relations without any desire to do so. When you see that you have no sex drive, you get very sad. It was really tough for me because I have been forced to have sexual relations on many occasions, and that’s really tough.” (P2, 63 years old, semi-structured interview), “I have always been an active woman, I have had fun with my husband like crazy, but after the illness, nothing. It really makes me mad, I get mad at myself. Before I handled it worse, now I don’t know if it’s because I am getting old, because of age, or, as the saying goes “love is running out because of using it so much”, and so it doesn’t affect me so much.” (P14, 69 years old, semi-structured interview)* Lack of professional consultation: *“No, it was never discussed. Now, any problem I have, I can talk about it with any professional, but I couldn’t do so back then. I should have sought help, but I didn’t do so… it could be because I was embarrassed, or because it didn’t occur to me to do so.” (P2, 63 years old, semi-structured interview)*
The catheter, the fistula and sexuality	Presence of devices which influence their sex life: *“Well, yes because this [peritoneal catheter] is something that I can’t hide. I talk with him about it, during sexual relations, I don’t even realize, but, sometimes, there are certain positions that perhaps I can’t do, because you think to yourself, ‘if I damage the catheter, after a while this rubbish doesn’t drain and I am going to have to go to the hospital’”. (P8, 32 years old, semi-structured interview), “I am unable to look, I see a hole there and cables [permanent venous catheter] and I can’t do it. I am trapped in it and I see it as something ugly.”* (*P10, 50 years old, semi-structured interview)* The fistula interferes the least with sexuality: “*The fistula has never interrupted anything, it hasn’t affected me at all. It conditions my partner more than myself, he told me to wear long sleeves.” (P2, 63 years old, semi-structured interview),* “*It isn’t easy to have a catheter [permanent venous catheter], because sometimes men want to stroke you and you have to say—‘hey, be careful, I have this’.” (P16, 42 years old, semi-structured interview)*.

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
