# Peer review of "Body Changes and Decreased Sexual Drive after Dialysis: A Qualitative Study on the Experiences of Women at an Ambulatory Dialysis Unit in Spain"

_ijerph, 2019, doi:10.3390/ijerph16173086_

Round 1
Reviewer 1 Report
My decision is accept
Reviewer 2 Report
I accept the publication of improved version
Reviewer 3 Report
My original suggestion was to reject this manuscript because it is a
rather descriptive than analytical paper. However, authors have
responded diligently and give a lot of revision. This deserves
encouragement. I agree that it can be considered to be accepted.
This manuscript is a resubmission of an earlier submission. The following is a list of the peer review reports and author responses from that submission.
Round 1
Reviewer 1 Report
The results were not presented in a scientific manner.
Why is it important to compare the changes in body and sexual functions among dialysis patients ? Why is this important in your population, in Spain?
Small population in study
Reviewer 2 Report
It is widely documented that CKD, particularly ESRD and dialysis treatment is associated with multiple depressive events, including libido diminishment and inferior self-evaluation. Your study confirms these observation, not adding novel information. Your group is small, with 10 persons above 50 years, age in which also in healthy population lower interest in sexual activity as well as body perception occurs. The issue of being on transplant waiting list and having prospect s for better life is omitted.
Reviewer 3 Report
The authors have chosen a difficult topic to study. The intention is worth encouraging, but the result did not come out as expected. Opinions are as followed.
Sexual activity or sexual life is complicated, involving many factors, physical as well as psychological. The interviewing of patients took long hours, which I admire, but there is one major neglected factor. How about their partners' perception and opinions? How can the sexual activity be performed without partners participation? Yes, they can masturbatem but I don't think that masturbation is included in this manuscript. Values would be added if the partners' opinions are also included.
This topic is less studied, but not unstudied. It's easy to find some publications on this topic. For example, PLoS One. 2017 Jun 20;12(6):e0179511 for dialoysis patients, and Transplant Proc. 2017 Nov;49(9):2099-2104. for transplantation patients.
This manuscript is more descriptive, but not analytical. Authors may consider other domain fields for publication.